

# How biased are our models? – A Case Study of the Alpine Region

Denise Degen[1], Cameron Spooner[2,3], Magdalena Scheck-Wenderoth[2,4], and Mauro Cacace[2]

[1]Computational Geoscience and Reservoir Engineering, RWTH Aachen University, Wüllnerstr. 2, 52062 Aachen, Germany
[2]Helmholtz Centre Potsdam GFZ German Research Centre for Geosciences, Telegrafenberg, 14473 Potsdam, Germany
[3]Institute of Earth and Environmental Science, Potsdam University, Potsdam, Germany
[4]Department of Geology, Geochemistry of Petroleum and Coal, RWTH Aachen University, Aachen, Germany

**Correspondence:** Cameron Spooner (spooner@gfz-potsdam.de)

**Abstract.** Geophysical process simulations play a crucial role in the understanding of the subsurface. This understanding is required to provide, for instance, clean energy sources such as geothermal energy. However, the calibration and validation of the physical models heavily rely on state measurements such as temperature. In this work, we demonstrate that focusing analyses purely on measurements introduces a high bias. This is illustrated through global sensitivity studies. The extensive

exploration of the parameter space becomes feasible through the construction of suitable surrogate models via the reduced basis method, where the bias is found to result from very unequal data distribution. We propose schemes to compensate for parts of this bias. However, the bias cannot be entirely compensated. Therefore, we demonstrate the consequences of this bias with the example of a model calibration.

## 1 Introduction

Understanding the subsurface is as important in the field of Geosciences as understanding climatic processes. In this paper, we focus on the understanding of the subsurface temperature field, which is of major importance for geothermal applications. Here, we focus on numerical process simulations to improve our understanding of the subsurface. These simulations are based on both geological and physical models, however in this paper, we will further investigate primarily the latter. The physical model has two major sources of uncertainties arising from the physical processes itself (i.e neglected processes, generalizations) (i.e.

Houghton et al., 2001; Murphy et al., 2004; Refsgaard et al., 2007) and from the physical parameters (i.e. thermal conductivity, radiogenic heat production) in terms of ranges (i.e. Freymark et al., 2017; Lehmann et al., 1998; Vogt et al., 2010; Wagner and Clauser, 2005) and their distribution (i.e. Feyen and Caers, 2006; Floris et al., 2001).

To compensate for both sources of uncertainties, one commonly performs model calibrations, either deterministically (i.e. Doherty and Hunt, 2010; Fuchs and Balling, 2016; Hill and Tiedeman, 2006; Wellmann and Reid, 2014) or stochastically (i.e.

Elison et al., 2019; Linde et al., 2017). Model calibrations aim to compensate for existing model error by adjusting the model parameters to a given data set. Naturally, the data set itself is subject to uncertainties. However, if we perform, for instance, stochastic model calibrations as Markov Chain Monte Carlo (Iglesias and Stuart, 2014), we are able to take these uncertainties into account. Nonetheless, there is another problem related to the data set and this is the data distribution. Note that in the





following, we introduce the problems arising from data distribution through the example of temperature measurements. Still,

many of the presented problems are generalizable for other geophysical data sources.

The first problem related to the data distribution is the depth location of the individual measurements. Our geothermal models have a depth in the magnitude of 100 km. In contrast, our deepest thermal measurements are commonly at a depth of 5 km to 7 km. The second problem is related to data density. Focusing on the horizontal data distribution, we face the problem of data sparsity and unequal data distribution. In certain model areas, we have very few temperature measurements and in other

areas, we have a much larger data density. This inequality can be compensated by using data weighting schemes (i.e. Degen et al., 2020a; Lerch, 1991). However, we also have areas where no temperature measurements exist. Data weighting cannot compensate for these non-existent measurements. The problem is further enlarged by the data source. Most of our temperature measurements come from the hydrocarbon industry, however, their targets and those of the geothermal industry are not the same in every region. This means that we can face the problem of lower data resolution in areas of interest whilst possessing

higher data resolution in areas that are not of primary interest.

The problem of data sparsity is long and widely recognized (i.e. Cherpeau and Caumon, 2015; Zehner et al., 2010). However, there are no studies systematically investigating the bias we introduce due to temperature measurements in a geothermal setting. Studies for the measurement bias are common in the field of remote sensing (i.e. Feng et al., 2016; Schwarz et al., 2020), however, their focus is entirely different. In remote sensing, the location of the measurements is subjected to uncertainties. In

contrast, our problems do not arising from imprecise measurement locations but their distribution. Naturally, our locations are also associated with uncertainties, however, in basin-scale applications they are of minor importance.

In this paper, we aim to provide a systematic investigation of the bias induced by measurement distribution. Therefore, we perform global sensitivity analyses to determine the influence of the model parameters (i.e. thermal conductivity, radiogenic heat production) on the model response (i.e. temperature). Sensitivity analyses can be subdivided into local and global analy-

ses. We choose a global sensitivity analysis to investigate not only the influence of the parameters itself but also the parameter correlations. Note that a local sensitivity analysis assumes that all parameters are independent of each other (Degen et al., 2020a; Saltelli, 2002; Saltelli et al., 2010; Sobol, 2001; Wainwright et al., 2014). Furthermore, we want to avoid a possible overestimation of the influences. A previous model study showed that the local sensitivity analysis can overestimate the influences (Degen et al., 2020a). Global sensitivity analyses have been performed before in, for example, Baroni and Tarantola

(2014); Cannavó (2012); Cloke et al. (2008); Degen et al. (2020a); Fernández et al. (2017); van Griensven et al. (2006); Song et al. (2015); Tang et al. (2007); Wainwright et al. (2014); Zhan et al. (2013), however, they are either in a different geophysical setting and or with a different focus of interest.

Global sensitivity analyses have the disadvantage of being computationally very demanding since they require several thousand to several hundred-thousands forward simulations. This makes these analyses infeasible even for state-of-the-art finite

element problems. To compensate for the expensive nature of the method, we employ the reduced basis method to construct suitable surrogate models. The principle idea is to replace the original high dimensional model with a low dimensional model while keeping the key characteristic of the problem (Benner et al., 2015; Hesthaven et al., 2016; Prud'homme et al., 2002; Quarteroni et al., 2015). In this paper, we do not focus on the observation space alone but also investigate the entire tempera-





ture state. Hence, we need a surrogate model for the entire state. The reduced basis method is able to provide us with this, in
contrast to many other surrogate model techniques (Baş and Boyacı, 2007; Bezerra et al., 2008; Frangos et al., 2010; Khuri and
Mukhopadhyay, 2010; Miao et al., 2019; Mo et al., 2019; Myers et al., 2016; Navarro et al., 2018). The reduced basis method
is widely known in mathematical applications (i.e. Benner et al., 2015; Grepl, 2005; Hesthaven et al., 2016; Aretz-Nellesen
et al., 2019; Kärcher et al., 2018; Prud'homme et al., 2002; Quarteroni et al., 2015; Rozza et al., 2007), however only few
geoscientific applications exist (Degen et al., 2020b). Nevertheless, some studies do use comparable approaches (Ghasemi and
Gildin, 2016; Gosses et al., 2018; Rizzo et al., 2017; Rousset et al., 2014; Zlotnik et al., 2015).

In this paper, we investigate the problems related to the data distribution for the case study of the Alpine Region. The
geological model, covering the Alpine orogen and its forelands, is taken from a previous study (Spooner et al., 2020). Thermal
studies of the Alpine Region are of interest to understand how the present-day deformation is linked to the thermal field.
Therefore, we want to illustrate how the interpretation of the temperature field might be biased.

## 2 Materials and Methods

In the following, we briefly introduce the concepts of global sensitivity analyses and the reduced basis method. Furthermore,
we introduce the physical model and the temperature data used throughout this study.

### 2.1 Global Sensitivity Analysis

In this study, we investigate the measurement bias and therefore require knowledge of which parameters the temperature
distribution is sensitive to. Therefore, we employ a sensitivity analysis (SA). We distinguish two types of sensitivity analyses:
local and global. The local sensitivity analysis investigates the influence of the model parameters with respect to a user-defined
reference parameter set. All parameter variations are considered independent of each other and only the vicinity of the input
parameters is explored (Sobol, 2001; Wainwright et al., 2014). In contrast, the global sensitivity analysis explores the entire
parameter space and also investigates the parameter correlations (Sobol, 2001). In this paper, we use a global sensitivity
analysis with the Saltelli sampler (Saltelli, 2002; Saltelli et al., 2010), and we investigate two types of sensitivity indices: the
first- and total order indices. First-order indices describe the influence arising from the model parameter itself. Total-order
indices additionally contain information about the parameter correlation (Sobol, 2001). We perform the SA with the Python
library SALib (Herman and Usher, 2017) and 100,000 realizations per parameter to reduce the statistical error. For further
information regarding the global sensitivity analysis refer to Sobol (2001); Saltelli (2002); Saltelli et al. (2010), and for a
comparison between local and global sensitivity analysis to Wainwright et al. (2014) and Degen et al. (2020a).





## 2.2 Forward Problem

For this case study, we are using a conductive heat transfer problem (Turcotte and Schubert, 2002). To ensure that we investigate the relative importance of the parameters and for better efficiency, we use the following non-dimensional form:

$$\frac{\lambda}{\lambda_{\text{ref}} \, S_{\text{ref}}} \, \frac{\nabla^2}{l_{\text{ref}}^2} \left( \frac{T - T_{\text{ref}}}{T_{\text{ref}}} \right) \; + \; \frac{S}{S_{\text{ref}} \, T_{\text{ref}} \, \lambda_{\text{ref}}} \; = \; 0, \tag{1}$$

where $\lambda$ is the thermal conductivity, $S$ the radiogenic heat production, and $T$ the temperature. The subscript "ref" denotes the respective reference parameters and $l_{\text{ref}}$ the reference length. Note that the Laplace-operator acts on the normalized space.

## 2.3 Reduced Order Modeling

In this work, we require a surrogate model that is representative of the entire temperature state to ensure the feasibility of the study. Therefore, we use the reduced basis (RB) method for the surrogate model construction, a projection based model
order reduction technique. It aims to replace the original high dimensional model with a low dimensional representation while keeping the input-output relationship the same. Hence, the method preserves the underlying physics. One limitation of the RB method is that it is restricted to underlying low dimensional parameter spaces. With higher dimensional parameter spaces the complexity of the parameter space tends to increase, leading to longer construction times and surrogate model dimensions that are too large. The RB method destroys the sparsity pattern of the system, meaning that a large surrogate model will
require a longer execution time than the original finite element model due to its dense nature. To overcome this issue, we use a hierarchical sensitivity study as we will discuss in Section 3.1.

The RB method compromises two parts: the offline and online stages. During the offline stage, we construct our surrogate model. This stage is computationally expensive but needs to be performed only once. In the online stage, we use the low-dimensional surrogate model. This stage is computationally fast and therefore ideal for expensive outer loop processes such
as the global sensitivity analysis. In previous studies, we showed that the RB method yields a speed-up of several orders of magnitude for the here described physical problem (Degen et al., 2020b, a).

All reduced models are generated with the software package DwarfElephant (Degen et al., 2020b). Degen et al. (2020b) also contains a detailed description of the reduced order model construction, which is omitted here for the sake of clarity. For further information regarding the RB method refer to Hesthaven et al. (2016); Prud'homme et al. (2002); Quarteroni
et al. (2015) and a detailed overview of various model order reduction techniques is provided in Benner et al. (2015). Further information regarding the RB method in the field of Geosciences is presented by Degen et al. (2020b) and specifically for basin-scale thermal applications in (Degen et al., 2020a).

## 2.4 Temperature Data

We present the temperature data set in form of a histogram in Fig. 1, and illustrate the spatial distribution in Fig 2. This
temperature data is identical to the one presented in Spooner et al. (2020). The entire data set compromises 8120 measurements with a maximum depth of 7.3 km and a mean depth of 1.8 km. The Italian National Geothermal Database (Trumpy and



Manzella, 2017) provides the data for the southern foreland. For the northern foreland, the data is derived from the Upper Rhine Graben data base provided in Freymark et al. (2017) and references therein. The data of the Molasse Basin is retrieved from Przybycin et al. (2015) and references therein, whereas the data from the Alps is compiled from Luijendijk et al. (2020).

The spatial distribution of measurements varies widely across the region, sparse in the Molasse Basin (103) and Alps (83) to dense in the Po Basin (7,619). In an effort to alleviate a significant bias and to improve the efficiency of the presented methods, the dataset was filtered to give a more uniform measurement density across the region, with a significant reduction in the Po Basin (2,028) whilst retaining those in the Molasse Basin (103) and Alps (83). Deeper measurements (> 2 km) were preferentially maintained throughout the region as they better indicate crustal temperatures, a particular focus of the work

undertaken here. This resulted in a filtered dataset of 2,388 wellbore temperatures measurements with a mean depth of 2.3 km.

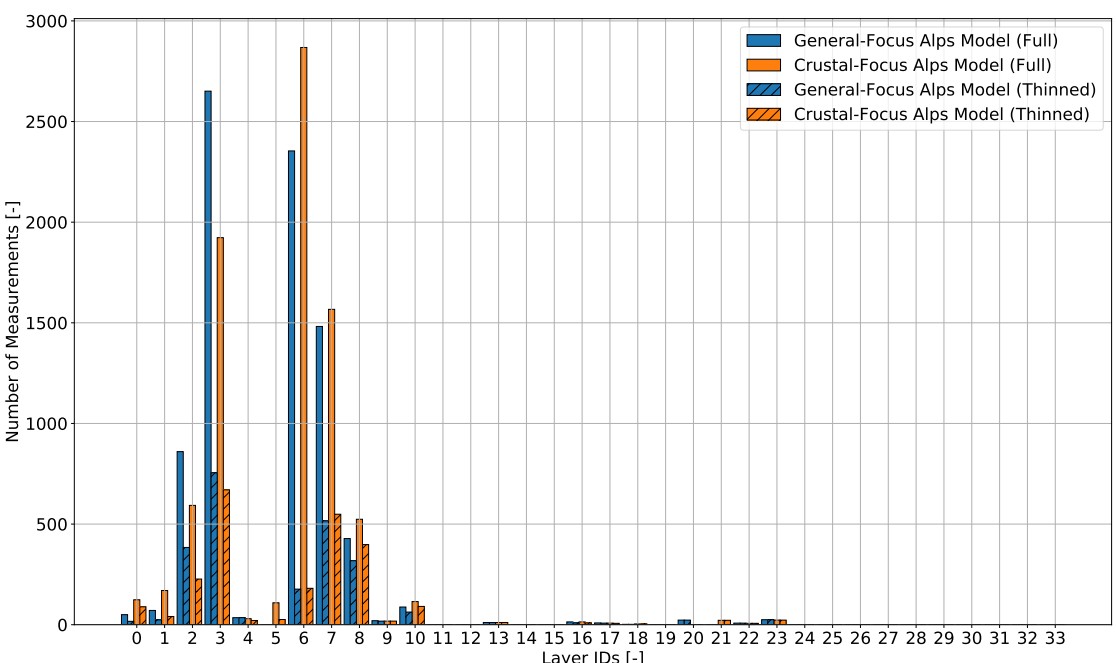

**Figure 1.** Distribution of the measurements according to the geological layers. For the Layer IDs please refer to Tab. A1.

### 2.4.1 Weighting

A common issue of the temperature data for the calibration of thermal models is their unequal distribution. To compensate for this inequality, we introduce a weighting scheme in this paper. There are different possibilities to weight the measurement data. In this paper, we use a regional weighting scheme that combines quantitative measures and our knowledge about the

geophysical settings and the data quality. As previously mentioned the data set was reduced to 2,388 data points in total. We subdivide the model into four regions:





**Figure 2.** Spatial distribution of the temperature measurements a) projected on the surface, b) along the crossection i, and c) along the crossection ii.

- – the Alps with 83 measurements,

- – the URG with 177 measurements,

- – the Molasse with 103 measurements,

- – and the Po Basin with 2028 measurements.

As we can see, the Po Basin contains many more temperature measurements than the other regions. Additionally, we need to take into account that the temperature measurements of the Alps are non-robust since they are minimum temperature values.





Also, the data from the Upper Rhine Graben needs to be treated carefully since we do not account for convective processes in this paper. These aspects yield the following weighting scheme:

- the Po Basin is not weighted,

- the Molasse is weighted by a factor of 20 since the Po Basin contains 20 times more data points,

- and the Upper Rhine Graben and the Alps are weighted by a factor 0.5.

## 3   Alpine Region

In this paper, we study two versions of the Alps Model:

1. The first one focuses on the Sediments and the Lithospheric Mantle. This model has been presented in Spooner et al. (2020) and is from here on denoted as the "General-Focus Alps" model. It consists of 31 geological layers. Each layer has a homogeneous and isotropic thermal conductivity and radiogenic heat production.

2. The second model concentrates on the Upper Crust and is denoted as the "Crustal-Focus Alps" model. This model contains 34 geological layers, again each layer has a homogeneous and isotropic thermal conductivity and radiogenic
heat production.

Both models have an extent of 640 km in the x-direction and 600 km in the y-direction. In the vertical direction, both models extend down to the Lithosphere-Asthenosphere Boundary (LAB). The models are discretized using hexaeders with a horizontal resolution of about 21.33 m × 19.35 m.

At the top of both models we apply a Dirichlet boundary condition representing the annual average surface temperatures
(Böhm et al., 2009; Fan and Van den Dool, 2008; Locarnini et al., 2013) varying from -10 °C (Alps) to 16 °C (Adriatic Sea). Additionally, at the base of the model, we assign a Dirichlet boundary condition varying between 1250 °C below the Vosges massif and 1400 °C below the Bohemian massif (Schaeffer and Lebedev, 2013). For further information regarding the physical and geological setting of the General-Focus Alps model refer to Spooner et al. (2020).

For the reference thermal conductivity, we use a value of 3.0 W m$^{-1}$ K$^{-1}$ (corresponding to the largest thermal conductivity).
Analogously, the reference length is 640,000 m (corresponding to the maximum model extent), and the reference radiogenic heat production 2.6 $\mu$W m$^{-3}$ (corresponding to the largest radiogenic heat production). The reference parameters are the same for both models.

In this paper, in addition to the General-Focus Alps model, already presented in Spooner et al. (2020) we use the Crustal-Focus Alps model, where the Upper Crust below the Po Basin was thinned in order to better fit temperature observations from
the previous thermal modeling work (Spooner et al., 2020), with requisite thickening of the Lower Crust carried out in order to compensate. Inconsistencies in the original classification of Unconsolidated Sediments and Consolidated Sediments were also rectified, specifically in the region of the Southern Alps. Small alterations to the depth of the Moho were also made as a result of more recent observations (Magrin and Rossi, 2020). The gravity residual of the newly generated structural model was then





re-minimised using the same methodology described in Spooner et al. (2019), achieving a misfit as good as the original model.

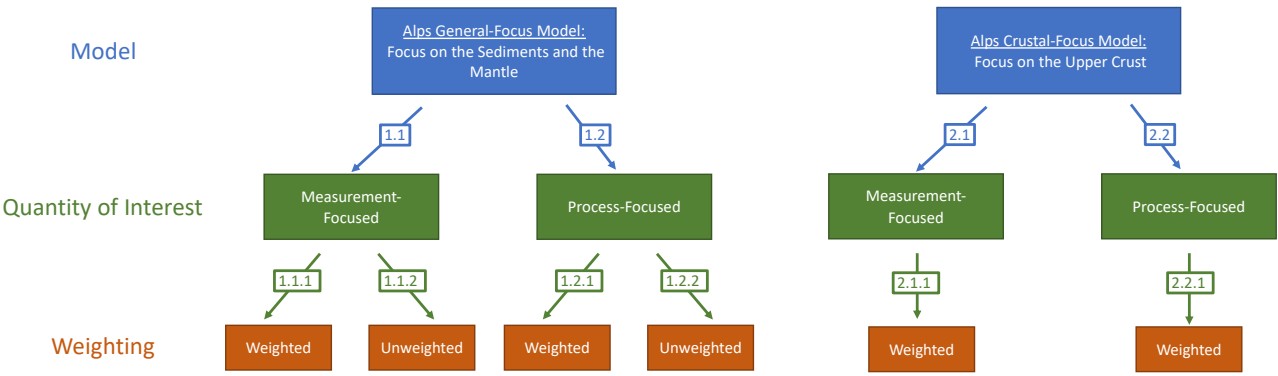

**Figure 3.** Schematic overview of the models used in this paper.


## 3.1 Thermal Model

To avoid the problem of the parameter space dimension becoming too large, we perform a hierarchical global sensitivity analysis. The setup for both the General-Focus and Crustal-Focus Alps model is the same. Therefore, we explain the hierarchical sensitivity analysis using the General-Focus Alps model. For the top level sensitivity analysis, we separately combine layers

with equal thermal conductivities and radiogenic heat productions, reducing the number of thermal parameters from 62 to 19. This top level sensitivity analysis investigates the influences of the thermal properties in the entire model region. However, the investigated properties combine several entities, so in order to isolate the thermal properties that are influencing the temperature distribution, we perform additional sensitivity analysis for those properties that exceed our threshold of $1 \cdot 10^{-2}$. This threshold was chosen at a level, where we observed a significant decrease in the sensitivity indices. In total, we perform three additional

sensitivity analysis for the:

1. Unconsolidated Sediments and the Lower Crust (red rectangle of Fig. 4 and Peak 1 of Fig. 5),

2. Unconsolidated and Consolidated Sediments (gray rectangle of Fig. 4 and Peak 2 of Fig. 5),

3. and the Upper Crust (blue rectangles of Fig. 4 and Peak 3 of Fig. 5).

Each of these additional sensitivity analyses also contains a thermal parameter from the top level sensitivity analysis to enable

a comparison between all analyses. We investigate all thermal properties of the Upper Crust and not only those that are above the threshold since the Upper Crust has been the primary interest in previous studies (Spooner et al., 2020). The setup of the hierarchical sensitivity analysis is shown in Fig. 4 and 5. Note that in this section we only present the setup of the hierarchical sensitivity analysis. A detailed presentation of the individual analyses follows in the next sections.



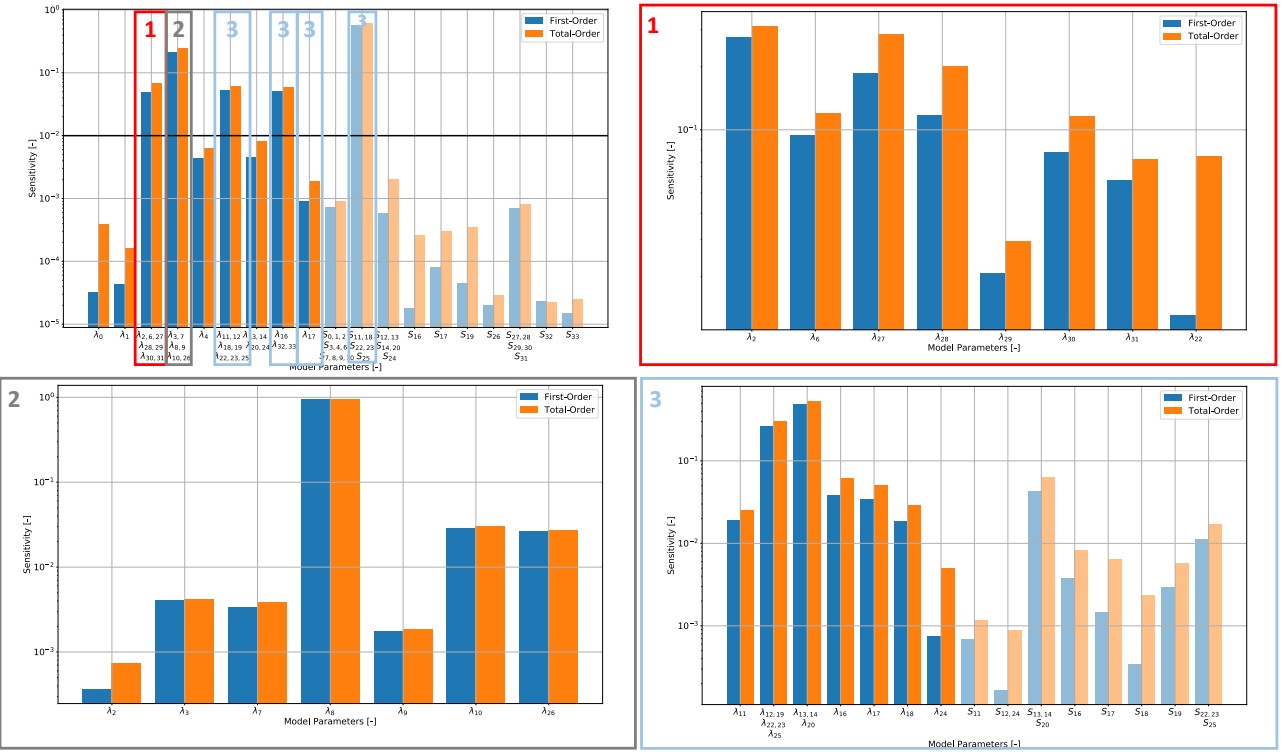

**Figure 4.** Representation of the hierarchical process-focused sensitivity analysis of the General-Focus Alps model. For the Layer IDs and acronyms please refer to Tab. A1.

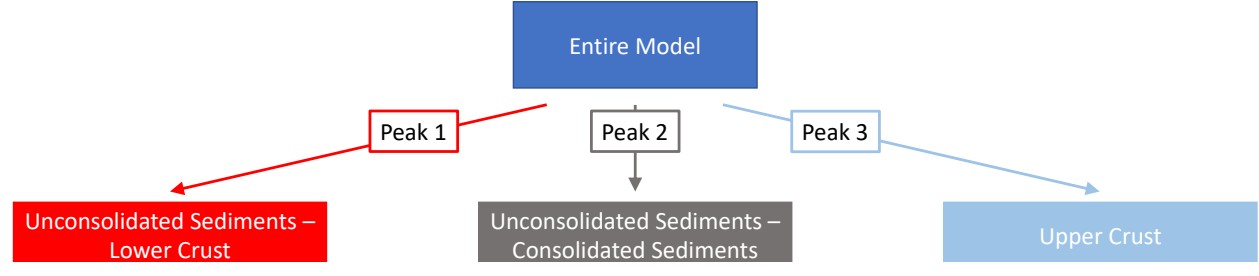

**Figure 5.** Schematic representation of the hierarchical global sensitivity analysis.

## 3.2 Influence of the Quantity of Interest

In this paper, we want to investigate how much our analyses are influenced by focusing on measurements. This is important since we calibrate and validate our analyses with, for instance, temperature measurements. The sensitivity analysis investigates the relative changes that are induced by changes in the model parameters (i.e. thermal conductivity and radiogenic heat production). For the sensitivity analysis, we need to define a quantity of interest, which allows us to define with respect to what





measure the changes are investigated. To investigate the influence of the measurements, we perform the hierarchical sensitivity
analyses with two different quantities of interest for the General-Focus Alps model (branch 1.1 and 1.2 of Fig. 3):

1. The first quantity of interest is defined as the sum of the absolute temperature values of the entire model. This results in
   a sensitivity analysis that is representative of the physical processes since all regions in the model are treated equally.

2. The second quantity of interest is defined as the absolute misfit between the simulated and measured temperature values.
   Hence, the resulting sensitivity analysis is focused on the temperature measurements.

In the following, we focus on the difference in the total order sensitivity indices between those two hierarchical sensitivity
analyses (branch 1.1 and 1.2 of Fig. 3) to present the bias introduced by the measurements and the consequences of using
temperature data from the hydrocarbon industry for the calibration of geothermal models. In this study, we use only the General-
Focus Alps model to avoid any influence from factors other than the measurements. Focusing on the difference between the

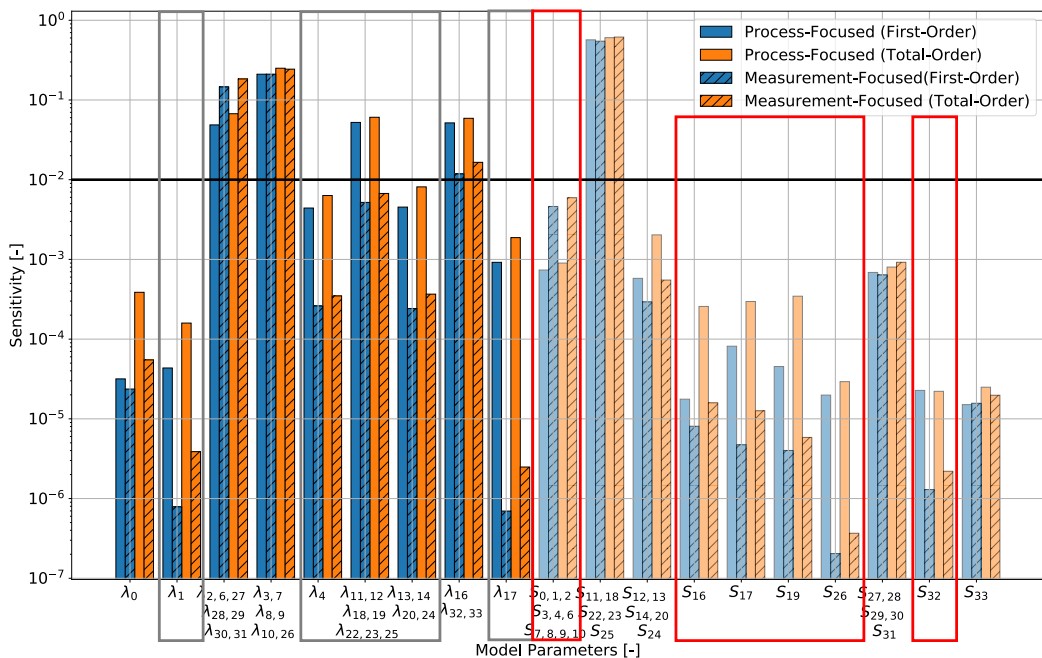

**Figure 6.** Top level sensitivity analysis (focusing on the entire Alps model) with different quantities of interest of the hierarchical global
sensitivity analysis for the General-Focus Alps model. For the Layer IDs and acronyms please refer to Tab. A1.

hierarchical sensitivity analyses, we make two key observations:

1. We observe tendentiously higher difference for the thermal conductivities of deeper geological layers. This is highlighted
   in Fig. 6 with gray rectangles. Here, we observe the highest differences for:





- $\lambda_1$ being the thermal conductivity of the Unconsolidated Sediments of the Upper Rhine Graben below 1 km,

- $\lambda_4$ being the thermal conductivity of the Unconsolidated Sediments of the Molasse Basin,

- $\lambda_{11}, \lambda_{12}, \lambda_{18}, \lambda_{19}, \lambda_{22}, \lambda_{23}$, and $\lambda_{25}$ compromising the thermal conductivities of the Appennine, Istrea, Molasse, East Alps, Po, and the North East and South East Adria Upper Crust,

- $\lambda_{13}, \lambda_{14}, \lambda_{20}$, and $\lambda_{24}$ compromising the thermal conductivities of the Moldanubia, Bohemia, West Alps, and Ivrea Upper Crust,

- $\lambda_{17}$ being the thermal conductivity of the Vosges Upper Crust.

Furthermore, this can be confirmed by looking at the additional sensitivity analysis of the Unconsolidated Sediments–Lower Crust (Fig. 7), where we observe higher differences for the Lower Crust thermal conductivities.

2. The difference in the sensitivity indices tend to be larger for the radiogenic heat production than for the thermal conductivity. This is highlighted in Fig. 6 and 9 with red rectangles.

Furthermore, in the case of the process-focused analyses, the model is sensitive to more parameters and we obtain a slightly higher parameter correlation.

Now, we focus on the difference observable for the analysis of the Unconsolidated and Consolidated Sediments. For both sediment types, we obtain huge differences in the sensitivities. For the thermal conductivities of the Unconsolidated Sediments, the measurement-focused analysis returns tendentiously higher influences, whereas for the Consolidated Sediments the process-focused analysis results in tendentiously higher influences of the thermal conductivities.

Finally, we switch our focus to the analysis of the Upper Crust. For the Upper Crust, we observe six thermal conductivities with a significant difference in the sensitivity indices:

- $\lambda_{13}, \lambda_{14}$, and $\lambda_{20}$ compromising the thermal conductivities of the Moldanubia, Bohemia, and West Alps Upper Crust,

- $\lambda_{16}$ being the thermal conductivity of the Saxothuringia Upper Crust,

- $\lambda_{17}$ being the thermal conductivity of the Vosges Upper Crust,

- $\lambda_{18}$ being the thermal conductivity of the Molasse Upper Crust,

- $\lambda_{24}$ and being the thermal conductivity of the Ivrea Upper Crust.

The differences for the radiogenic heat production are the highest for:

- $S_{12}$ and $S_{24}$ compromising the radiogenic heat production of the Istrea and Ivrea Upper Crust,

- $S_{22}, S_{23}$, and $S_{25}$ compromising the radiogenic heat production of the Po, North East, and South East Adria Upper Crust.

Note that we do not present the results of the Upper Crust sensitivities in Fig. 7 and the Lower Crust sensitivities in Fig. 8. Both are properties from the top level sensitivity analysis and are required to enable a comparison between the different analyses. However, they represent only one property from the lithological unit. Therefore, they are not representative for any kind of trend analysis.



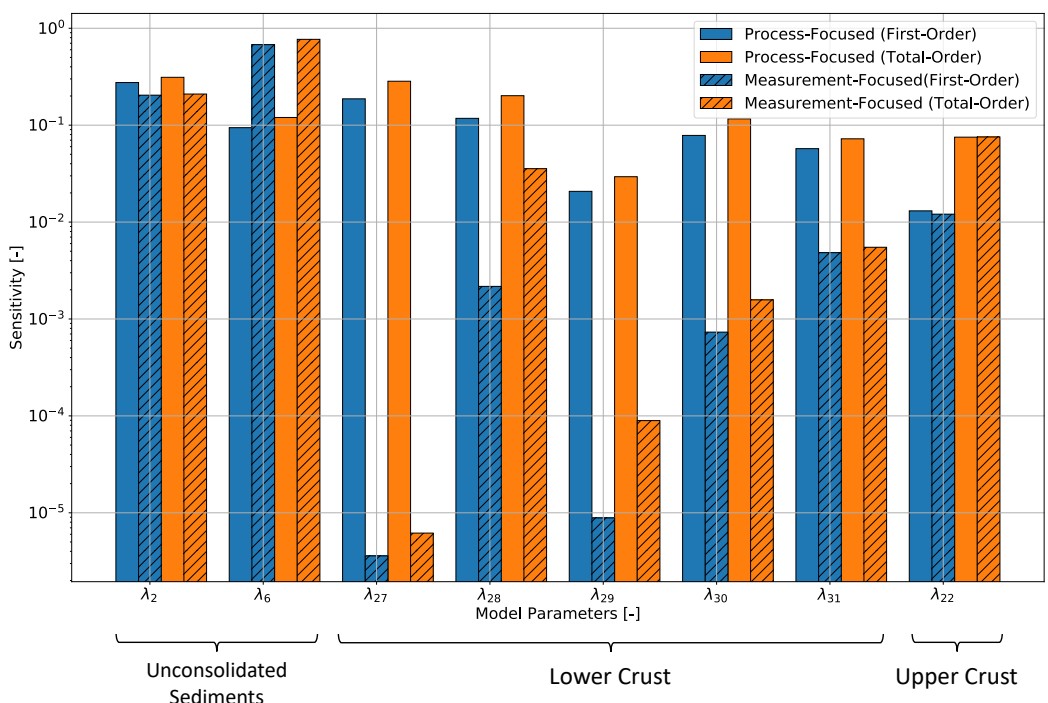

**Figure 7.** Sensitivity analysis of the Unconsolidated Sediments and Lower Crust with different quantities of interest of the hierarchical global sensitivity analysis for the General-Focus Alps model. For the Layer IDs and acronyms please refer to Tab. A1.

## 3.3 Influence of the Weighting

The consequences of introducing a weighting scheme have been already partly addressed in Degen et al. (2020a). However, there the authors focused on the consequences for the process of model calibrations. Here, we want to investigate how we can compensate for the measurement bias by applying weights.

Analogous to the previous section, we focus on the differences in the total order sensitivity indices. For all analyses, we can observe that the weighted scenario tends to be closer to the process-focused analysis than the non-weighted scenario for the thermal conductivities. This is highlighted by the gray rectangles in Fig. 10 and 13. The behavior is very prominent for the thermal conductivity of the Moldanubia Lower Crust (gray rectangle of Fig. 11).

In contrast, we observe for the thermal conductivities of the Upper Rhine Graben layers a closer resemblance of the non-weighted scenario to the process-focused analysis (blue rectangle of Fig. 10).

We also observe, for the radiogenic heat production, that for most layers the indices of the weighted case are closer to the process-focused analysis than the non-weighted (red rectangles of Fig. 10). Differing from this trend is the radiogenic heat production of the Istrea and Ivrea Upper Crust. Furthermore, we observe that the weighted analysis overestimates the influence of the Molasse Upper Crust (Fig. 13).



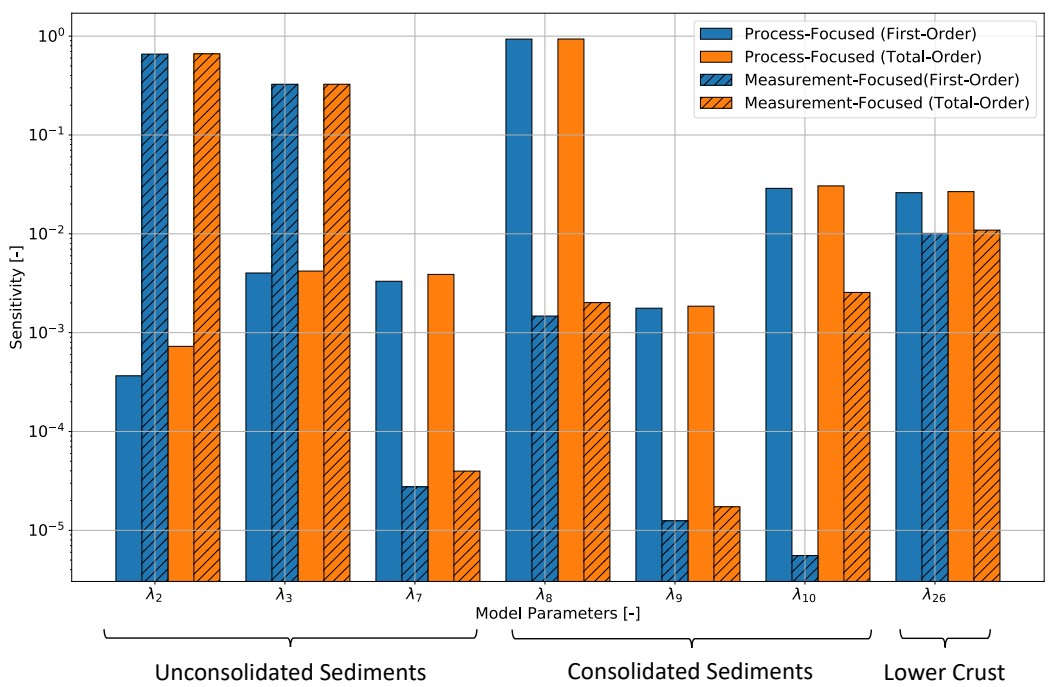

**Figure 8.** Sensitivity analysis of the Unconsolidated and Consolidated Sediments with different quantities of interest of the hierarchical global sensitivity analysis for the General-Focus Alps model. For the Layer IDs and acronyms please refer to Tab. A1.

## 4 Discussion

In the following, we discuss the consequences of focusing a study on measurements. Therefore, we discuss the changes in the sensitivities for the different quantities of interest and weighting schemes. Furthermore, we demonstrate the consequences

through a deterministic model calibration example.

### 4.1 Influence of the Quantity of Interest

The different quantities of interest represent the bias introduced by the unequal distribution of the measurement locations. Hence, we can use the difference in the sensitivity analysis to discuss the bias that is induced by the temperature measurements. So far, we had two key observations for the study of the different quantities of interest:

1. the difference in the indices for the thermal conductivities are higher for deeper layers,

2. the differences are higher for the radiogenic heat productions than for the thermal conductivities.

Both of these observations can be explained by having a closer look at the depth distribution of the temperature measurements (Fig. 14). We can see that most measurements are located in a depth of up to 2 km. The deepest measurement is at depth of about 7.3 km, whereas the model extends to a maximum depth of about 140.5 km. Hence, most measurements are located in



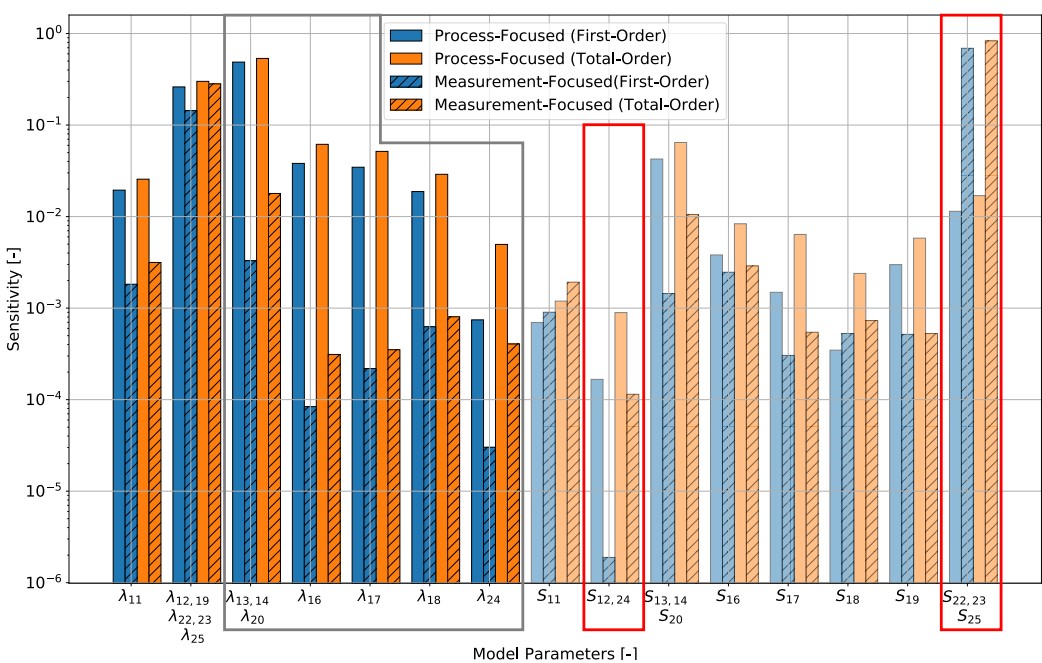

**Figure 9.** Sensitivity analysis of the Upper Crust with different quantities of interest of the hierarchical global sensitivity analysis for the General-Focus Alps model. For the Layer IDs and acronyms please refer to Tab. A1.

shallower geological layers, and in the deepest layers, we find no measurements at all (Fig. 1). Therefore, the measurement-focused analysis tends to underestimate the influences of the deeper geological layers and overestimates the influences of shallower. This is true for both thermal conductivity and radiogenic heat production.

We investigate the phenomenon closer for the analysis of the Unconsolidated and Consolidated Sediments. Here, we have a prominent overestimation of the influences of the Unconsolidated Sediments and an underestimation of the Consolidated

Sediments. We have:

- 384 data points in the Unconsolidated Sediments of the Upper Rhine Graben above 1 km ($\lambda_0$ in Fig. 6),

- 755 data points in the Unconsolidated Sediments of the Upper Rhine Graben below 1 km ($\lambda_1$ in Fig. 6),

- 516 data points in the Unconsolidated Sediments of the Po Basin below 2 km ($\lambda_7$ in Fig. 6),

- 318 data points in the Consolidated Sediments outside of sedimentary basins ($\lambda_8$ in Fig. 6),

- 18 data points in the Consolidated Sediments of the Molasse Basin ($\lambda_9$ in Fig. 6),

- and 63 data points in the Consolidated Sediments of the Po Basin ($\lambda_{10}$ in Fig. 6).





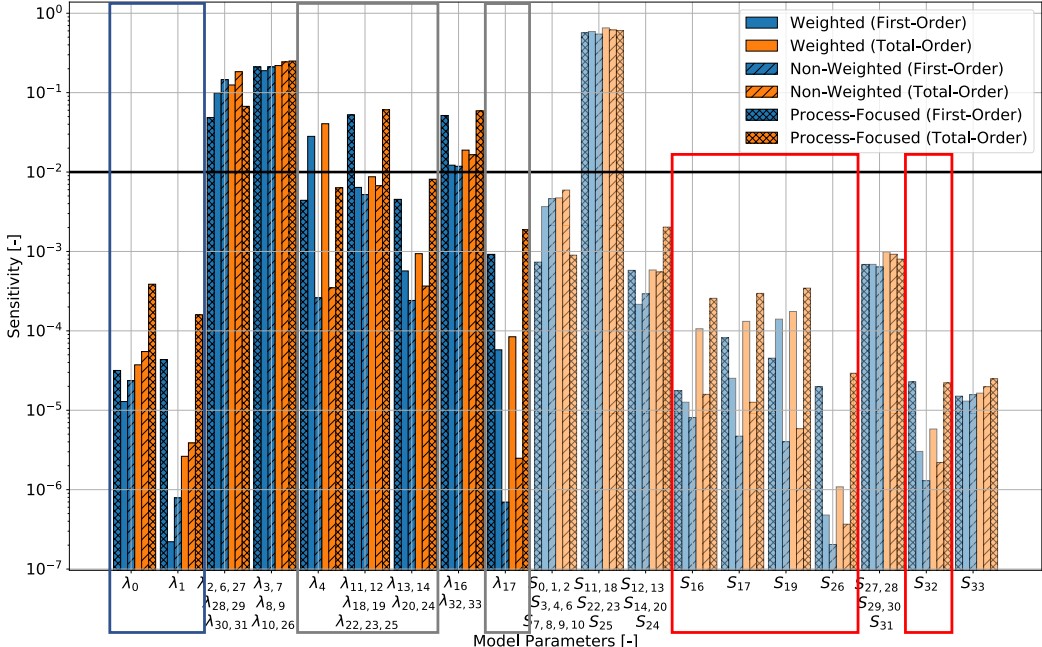

**Figure 10.** Top level sensitivity analysis (focusing on the entire Alps model) with different weighting schemes of the hierarchical global sensitivity analysis for the General-Focus Alps model. For the Layer IDs and acronyms please refer to Tab. A1.

The much higher data density in the Unconsolidated Sediments explains the high influence of the thermal conductivities of the Unconsolidated Sediments for the measurement-focused analysis. The only remaining question is why the influence of the thermal conductivity of the Unconsolidated Sediments Po below two kilometers is underestimated although containing 516 data points. This might be a bias introduced by the high data density of 755 data points in the Unconsolidated Sediments ($\lambda_3$).

The behavior is more pronounced for the radiogenic heat production for lithological reasons. The highest influences of the radiogenic heat productions arise from the Upper Crust (Fig. 6), meaning that the radiogenic heat production is more prominent in deeper parts of the model. However, these parts of the model are further away from our measurement locations. Hence, the measurement-focused analysis highly underestimates the influence of the radiogenic heat production. The same effect can be observed for the thermal conductivity of the Upper Crust ($\lambda_5$ in Fig. 6). For the measurement-focused analysis, the influence of the thermal conductivity is below the threshold, whereas for the process-focused analysis it is above.

The consequence of the data distribution becomes obvious once we look at the analysis of the Unconsolidated Sediments and Lower Crust (Fig. 7). For all lower crustal layers, the influence is significantly underestimated in the measurement-focused scenario. Consequently, by focusing on the measurement in the further analysis we would lose all information related to the Lower Crust, although the layer might be important for the physical understanding of the subsurface.



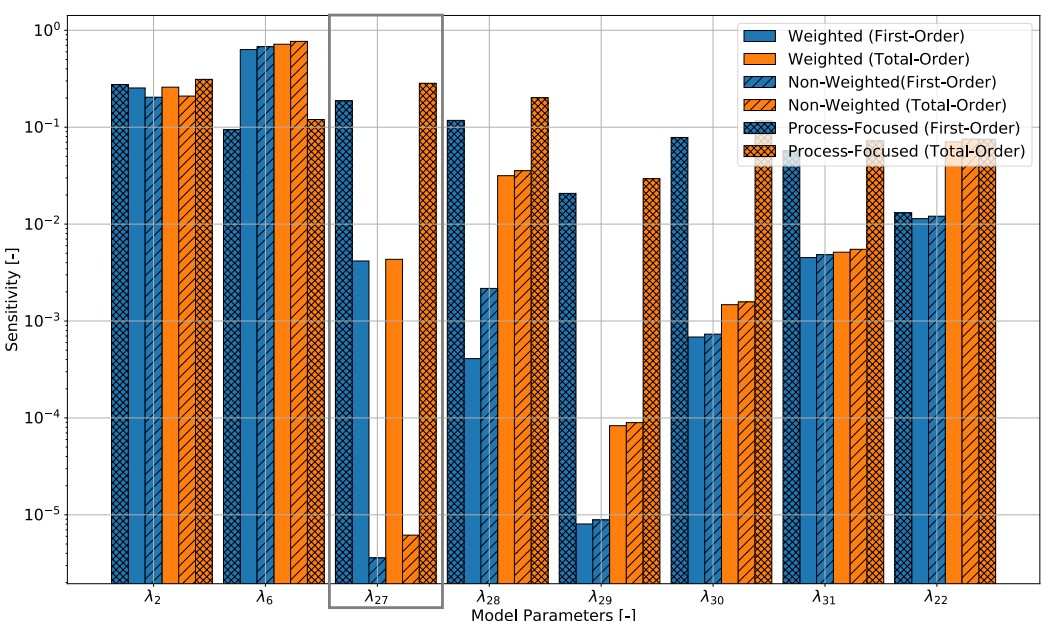

**Figure 11.** Sensitivity analysis of the Unconsolidated Sediments and Lower Crust with different weighting schemes of the hierarchical global sensitivity analysis for the General-Focus Alps model. For the Layer IDs and acronyms please refer to Tab. A1.

Also, for the analysis of the Upper Crust (Fig. 9), we are confronted with the consequences of the unequal data distribution. The huge difference in the influences of the thermal conductivities of the Saxothuringia, Vosges, Molasse, and Ivrea Upper Crust is caused by a very low or zero data density. Also, the influence of the Moldanubia, Bohemia, and West Alps Upper Crust is underestimated. We have data in the Moldanubia and West Alps Upper Crust but no data in the Bohemia Upper Crust yielding this discrepancy.

The influence of the radiogenic heat production of the Istrea and Ivrea Upper Crust is underestimated in the measurement-focused study due to the lack of data. Whereas the influence of the radiogenic heat production of the Po, North East Adria, and South East Adria Upper Crust is overestimated. This is likely caused by the measurements available for both the Po and North East Adria Upper Crust layers.

We also observed slightly higher parameter correlations for the process-focused analysis. This is probably related to the fact that the model is sensitive to more parameters.

## 4.2 Influence of the Weighting

We observed that the weighted measurement-focused analysis tends to be closer to the process-focused analysis. This becomes understandable by looking at the applied weighting scheme. We applied a regional weighting scheme to compensate for the unequal data distribution in the four regions of our model. Hence, we can compensate partly for the measurement bias. However, we are not able to fully compensate for the data sparsity. The main reason for this is that we can compensate for fewer data




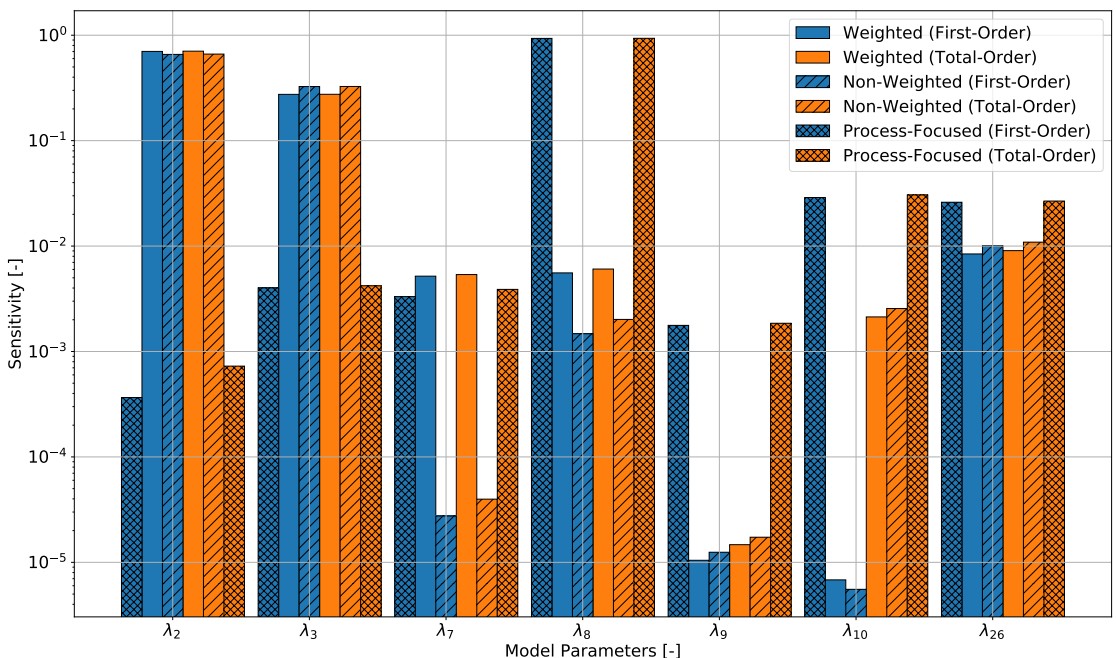

**Figure 12.** Sensitivity analysis of the Unconsolidated and Consolidated Sediments with different weighting schemes of the hierarchical global sensitivity analysis for the General-Focus Alps model. For the Layer IDs and acronyms please refer to Tab. A1.

points but not for regions without data points since no measurements are available to which we could apply a higher weight. This can be observed, for instance, in the properties related to the layers of the Molasse.

We observed that the sensitivity indices of the thermal properties related to the layers inside the Upper Rhine Graben are further apart for the weighted and process-focused comparison than for the non-weighted process-focused one. This is related to the choice of the weighting scheme. We chose to put less weight on the temperature data from the Upper Rhine Graben since we do not account for convective effects in this paper. Analogously, the properties of the Apennine Upper Crust layers also have a too small influence for the weighted scenario. As a reminder, we downgraded the importance of the temperature data in

this region since the data consists of minimum temperature data.

Through the weighting we are able to compensate for the underestimation of the Unconsolidated Sediments of the Po Basin. Hence, the bias most likely induced by the high data density of the other layers can be removed.

For the thermal conductivities of the Saxothuringia, Vosges, Molasse Upper Crust (gray rectangle of Fig. 13), we are again able to remove parts of the data bias caused by the data sparsity of these layers. The same phenomenon is observable for the

radiogenic heat production of the Upper Crust (red rectangles of Fig. 13).



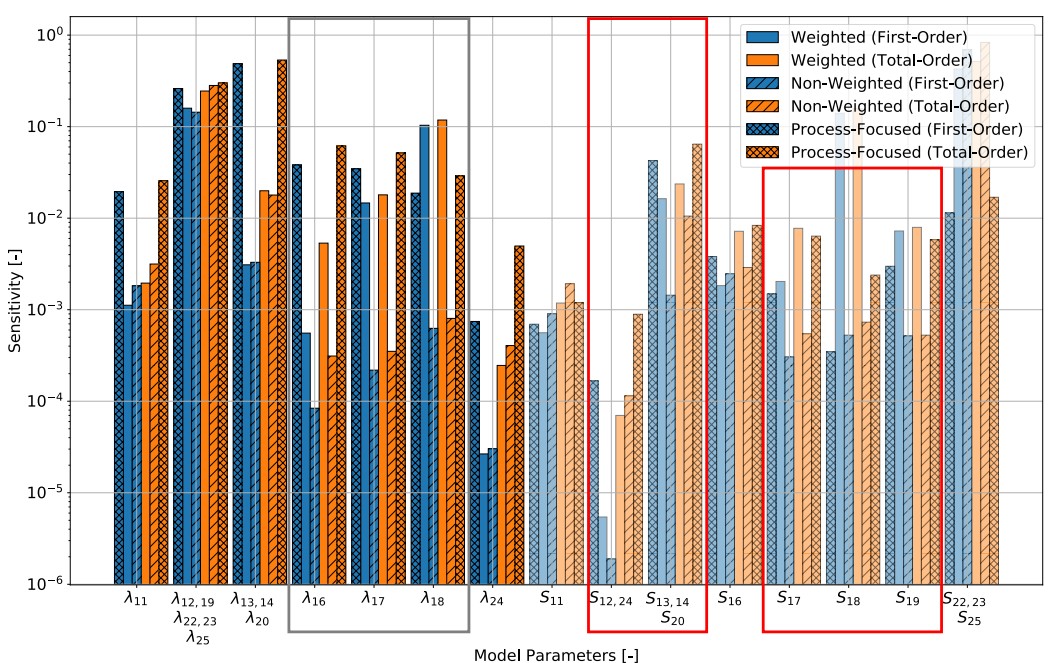

**Figure 13.** Sensitivity analysis of the Upper Crust with different weighting schemes of the hierarchical global sensitivity analysis for the General-Focus Alps model. For the Layer IDs and acronyms please refer to Tab. A1.

Note that the weighting scheme is case study and aim specific. Depending on our knowledge about data quality, regions of interest, and other aspects the weighting scheme can be case-specifically designed. In this paper, we do not aim to provide "the ideal" weighting scheme for the Alpine Region. Instead, we demonstrate the impact of a weighting scheme for thermal modeling.

### 4.3 Calibration Example

So far, we have presented that we obtain significantly differing sensitivities for the process-focused and measurement-focused study. In the following, we demonstrate the consequences of this difference through a deterministic model calibration. We choose the example of a model calibration because this is a typical inverse process that relies on observation data.

Model calibration aims to compensate for existing model errors by adjusting the model parameters in accordance with our temperature measurements. Analogous to Degen et al. (2020a), we use a sensitivity-driven model calibration for more robust results. In this study, we performed various sensitivity analyses. For the model calibration, we require the measurement-focused sensitivity analyses (branch 1.1 and 2.1 of Fig 3). We need these sensitivity analyses because they represent the information content that can be derived from the temperature data. In the case of the General-Focus Model, five thermal parameters that



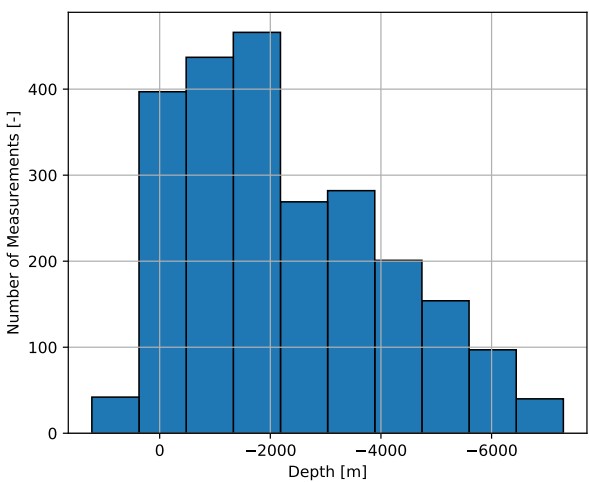

**Figure 14.** Distribution of the measurements according to the depth.

can be calibrated are yielded (Table 1). The data is insensitive to the remaining parameters. Hence, we cannot calibrate these

values. We are left with mostly shallow layers to calibrate. The exception is the Lithospheric Mantle which is influencial due to its large volume.

In the following, we discuss the results of the automated model calibration and its consequences. Note that in this work, we use the model calibration in a slightly different way. Usually, it is used to compensate for model errors. That means of course that it also identifies the problematic model areas. In this work, we employ the model calibration as an identification tool

for model errors. Therefore, we use as initial values the calibrated values by Spooner et al. (2020), which have been obtained through a "trial-and-error" model calibration. Then, large discrepancies between our initial values and calibrated values identify model problems.

The first model problem that we can identify is the measurement bias through an unequal data distribution (General-Focus – Unweighted). This can be at least partly removed through data weighting (General-Focus – Weighted) yielding smaller

differences between initial and calibrated values. Nonetheless, we observe a low radiogenic heat production of the Upper Crust, meaning that our model is non-ideal in the description of the Upper Crust. This also leads to thermal conductivities that are too low in the Sediments and too high in the Lithospheric Mantle.

Therefore, we introduce a second model, the Crustal-Focus Model. For this model, we obtain a good agreement for the Upper Crust but greater discrepancies in Unconsolidated Sediments (below 1 km) and the Lithospheric Mantle. Hence, we can

remove the error of the Upper Crust but at the same time introduce new error sources.

Note that we do not aim to present the "optimal" model in this paper. Instead, we want to demonstrate various components that influence the model. Generating an optimal model is not possible since all models are per definition wrong (Box, 1979).





**Table 1.** Comparison of the initial thermal properties and the calibrated thermal properties for different geological models and different weighting schemes.

| Parameter | Initial Value | Calibrated Value | | |
|---|---|---|---|---|
| | | General-Focus – Unweighted | General-Focus – Weighted | Crustal-Focus – Weighted |
| $\lambda_2$ [W m$^{-1}$ K$^{-1}$] | 2.0 | 1.53 | 1.70 | 2.02 |
| $\lambda_3$ [W m$^{-1}$ K$^{-1}$] | 2.3 | 1.33 | 2.04 | 3.45 |
| $\lambda_{4,5}$ [W m$^{-1}$ K$^{-1}$] | 1.8 | n/a | 1.62 | 1.53 |
| $\lambda_6$ [W m$^{-1}$ K$^{-1}$] | 2.0 | 1.86 | 2.02 | 2.03 |
| $\lambda_{32,33}$ [W m$^{-1}$ K$^{-1}$] | 3.0 | 3.71 | 3.18 | 2.5 |
| $S_{22,23,25}$ [$\mu$W m$^3$] | 1.3 | 0.2 | 0.8 | 1.3 |

We present here two models that fulfill different purposes. The General-Focus Model is better if we are interested in the entire model domain. In the case that our area of interest is only the Upper Crust, the Crustal-Focus Model is preferable.

## 4.4 Influence of the Model

We have discussed the consequences of the model change for the calibrated thermal conductivities. Now, we want to briefly discuss the consequences for the sensitivities. Therefore, we repeat the process-focused and measurement-focused sensitivity analysis for the Crustal-Focused model. Note that we consider only the weighted scenario (branch 2.1.1 and 2.2.1 of Fig. 3).

For the Crustal-Focused model, we thinned the Upper Crust. This can be clearly observed, in the decreased sensitivities of the model to the Upper Crust layers (red box of Fig. 15). However, this change is only visible in the process-focused analysis. The measurement-focused analysis mostly fails to resolve these changes due to the data sparsity in the Upper Crust (red box of Fig. 16). Underestimated changes are observable for the Saxothuringia Upper Crust. This highlights again the information loss of measurement-focused studies and the dangers associated with calibrations.

The radiogenic heat production of the most of the Lower Crust is more influencial for the Crustal-Focused model since the Upper Crust was thinned by thickening the Lower Crust. The only exception is the Saxothuringia Lower Crust ($\lambda_{26}$). For the process-focused analysis (Fig. 15) it loses importance and for the measurement-focused analysis (Fig. 16) it gains importance. For both models, we apply a Dirichlet boundary condition at the top and the bottom of the model. Hence, the temperature distribution is determined by the ratio of the thermal properties. Therefore, the difference in the Saxothuringia Lower Crust likely arising from the changes of other geological layers. The same is likely for the changes of the thermal conductivity of the Unconsolidated Sediments in the Molasse Basin. Also, the changes of the influences arising from the radiogenic heat production of the Lithospheric Mantle are caused by other layers, especially considering the very low values of these layers.

Furthermore, we observer a higher influence of the Unconsolidated Sediments in the Upper Rhine Graben (gray box of Fig. 15) although the model has not been changed around the Upper Rhine Graben. However, this might be an effect of the reclassification in the Unconsolidated and Consolidated Sediments. These changes are more pronounced for the measurement-



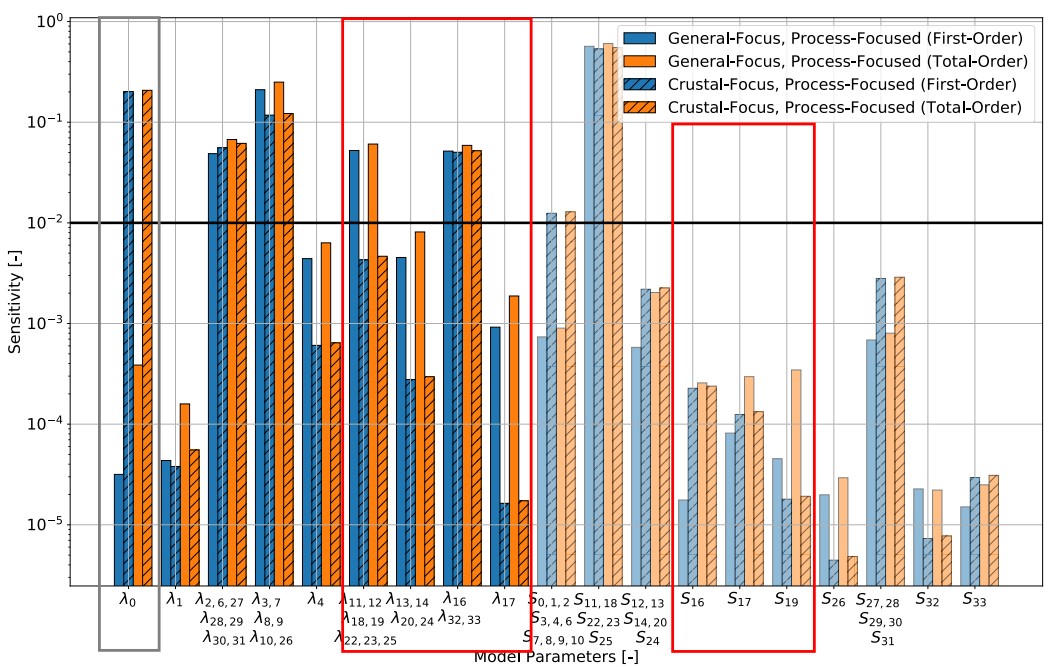

**Figure 15.** Comparison of the sensitivities of the process-focused study for both the General-Focus and Crustal-Focus Alps Model.

focused (gray box of Fig. 16) than for the process-focused analysis. This is again caused by the data distribution since we have more measurements at a shallower depth.

## 4.5 Gravity Model

The model change is observable in both the model calibration for the thermal properties and the corresponding sensitivities. However, if we look at the gravity residuals (Fig. 17), we do not observe any significant changes. This highlights a general point for the construction of geological models. We have different data sources available for the construction of a geological model. It is crucial to incorporate multiple data sources and not rely on a single data source. If we would have constructed a model of the Alps purely based on gravity, we would not have been able to identify the problem of the thickness of the Upper Crust.

## 4.6 Outlook

In this paper, we have seen that the measurements induced a significant bias. This opens the discussion of subsequent projects. Therefore, we would like to investigate how we can decrease this bias by incorporating further data sources that give us only



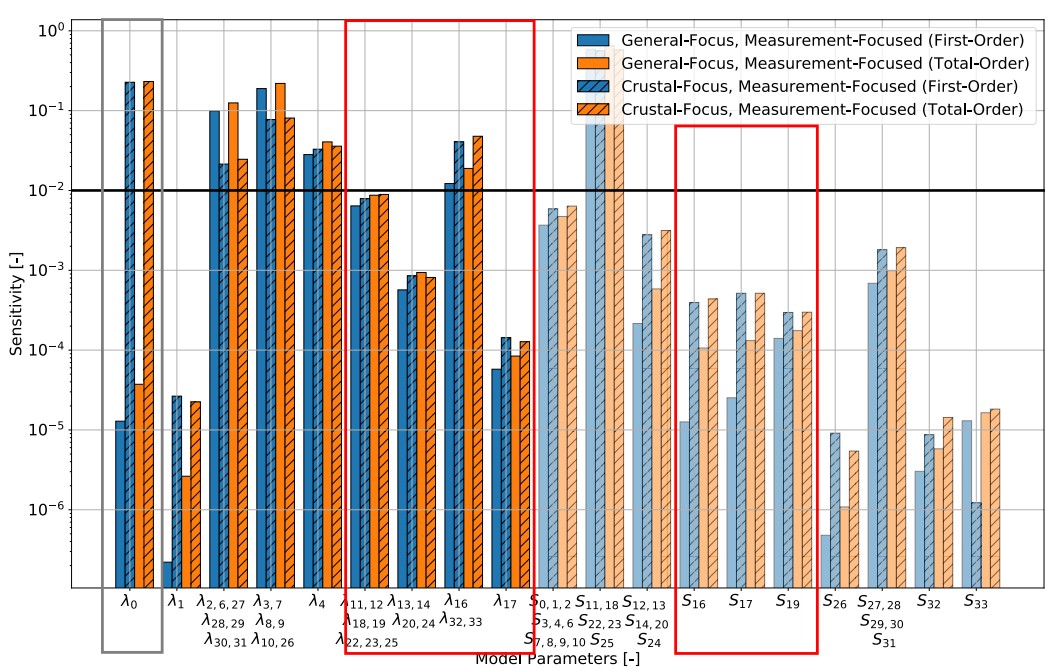

**Figure 16.** Comparison of the sensitivities of the measurement-focused study for both the General-Focus and Crustal-Focus Alps Model.

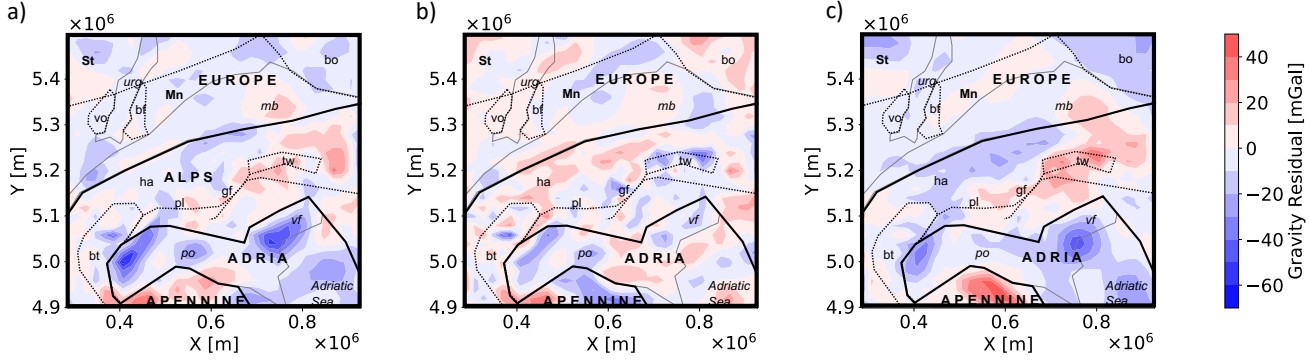

**Figure 17.** Gravity residual of a) the General-Focus Model, b) the Crustal Focus Model, and c) the difference between the General-Focus and Crustal-Focus Model. Acronyms - St - Saxothuringian Zone; Mn – Moldanubian Zone; Ha – Helvetic Alps; bo – Bohemian Massif; vo – Vosges Massif; bf – Black Forest Massif; tw –Tauern Window; bt – Briançonnais Terrane; pl – Periadriatic Lineament; gf – Guidicarie Fault; urg – Upper Rhine Graben; mb – Molasse Basin; po – Po Basin; vf – Veneto–Friuli plain.



an indirect measure of the temperature. Furthermore, it would be interesting to further explore the field of joint inversion to incorporate various geophysical data sources already used during model construction.

## 5 Conclusion

Throughout the entire paper, we have demonstrated the bias that a measurement-focused study can cause. This bias can be partly removed through automated and customized data weighting schemes. However, as typical for geoscientific applications, many areas of the model do not have any associated data. Unfortunately, it is not possible to compensate for the bias arising from these areas. This shows the importance of focussing on regions where data is present, whenever possible.

However many inverse processes such as deterministic and stochastic model calibrations are dependent on measurement

data. Here, this bias is unavoidable. Nonetheless, we need to be aware of which kind of bias we are introducing through this procedure to take the effects for all further analyses into account. We need to be aware that the data is often only informative towards the shallower layers. Hence, we lose the information about deeper layers and at the same time overestimate the influence of the shallower layers. This also means that we are unable to calibrate and validate the lower parts of our geological models. Nonetheless, these parts are important to avoid influences from, for instance, the lower boundary condition.

We have also seen the importance of considering various data sources. The model changes from the General-Focus to the Crustal-Focus model were only visible in the thermal studies but not in the gravity residuals.

Note that although we performed the analyses for the case study of the Alps these aspects hold in general since the data distribution shown here is typical for geoscientific applications.

*Code availability.* For the construction of the reduced models, we used the software package DwarfElephant (Degen et al., 2020b, c). The

software, which is based on the finite element solver MOOSE (Alger et al., 2019), is freely available on Zenodo (https://zenodo.org/badge/latestdoi/117989215). The sensitivity analyses are performed with the Python library SALib (Herman and Usher, 2017) and the model calibrations with the Python library scipy (Jones et al., 2014).

*Data availability.* The structural model 3D-ALPS, constrained in Spooner et al. (2019), used in the General-Focus Model is freely available as DOI and online material via the following link https://doi.org/10.5880/GFZ.4.5.2019.004. The thermal field (3D-ALPS_TR), generated

in Spooner et al. (2020), used in the General-Focus Model is freely available as DOI and online material via the following link https://doi.org/10.5880/GFZ.4.5.2020.007.

## 1 Appendix Acronyms and Layer IDs



Table A1: Acronyms and Layer IDs for both the General-Focus and Crustal-Focus Model

| Layer | Layer ID | Property | Acronym |
|---|---|---|---|
| Unconsolidated Sediments URG (top 1 km) | 0 | thermal conductivity | $\lambda_0$ |
| | | radiogenic heat production | $S_0$ |
| Unconsolidated Sediments URG (below 1 km) | 1 | thermal conductivity | $\lambda_1$ |
| | | radiogenic heat production | $S_1$ |
| Unconsolidated Sediments Rest (top 1 km) | 2 | thermal conductivity | $\lambda_2$ |
| | | radiogenic heat production | $S_2$ |
| Unconsolidated Sediments Rest (below 1 km) | 3 | thermal conductivity | $\lambda_3$ |
| | | radiogenic heat production | $S_3$ |
| **General-Focus Model:** | | | |
| Unconsolidated Sediments Molasse | 4 | thermal conductivity | $\lambda_4$ |
| | | radiogenic heat production | $S_4$ |
| **Crustal-Focused Model:** | | | |
| Unconsolidated Sediments Molasse (top 1 km) | 4 | thermal conductivity | $\lambda_4$ |
| | | radiogenic heat production | $S_4$ |
| Unconsolidated Sediments Molasse (below 1 km) | 5 | thermal conductivity | $\lambda_4$ |
| | | radiogenic heat production | $S_4$ |
| Unconsolidated Sediments Po Basin (top 2 km) | 6 | thermal conductivity | $\lambda_6$ |
| | | radiogenic heat production | $S_6$ |
| Unconsolidated Sediments Po Basin (below 2 km) | 7 | thermal conductivity | $\lambda_7$ |
| | | radiogenic heat production | $S_7$ |
| Consolidated Sediments | 8 | thermal conductivity | $\lambda_8$ |
| | | radiogenic heat production | $S_8$ |
| Consolidated Sediments Molasse | 9 | thermal conductivity | $\lambda_9$ |
| | | radiogenic heat production | $S_9$ |
| Consolidated Sediments Po Basin | 10 | thermal conductivity | $\lambda_{10}$ |
| | | radiogenic heat production | $S_{10}$ |
| Upper Crust Apennine | 11 | thermal conductivity | $\lambda_{11}$ |
| | | radiogenic heat production | $S_{11}$ |
| Upper Crust Istrea | 12 | thermal conductivity | $\lambda_{12}$ |
| | | radiogenic heat production | $S_{12}$ |
| Upper Crust Moldanubia | 13 | thermal conductivity | $\lambda_{13}$ |





| | | radiogenic heat production | $S_{13}$ |
|---|---|---|---|
| **General-Focus Model:** | | | |
| Upper Crust Bohemia | 14 | thermal conductivity | $\lambda_{14}$ |
| | | radiogenic heat production | $S_{14}$ |
| **Crustal-Focused Model:** | | | |
| Upper Crust Bohemia | 14 | thermal conductivity | $\lambda_{14}$ |
| | | radiogenic heat production | $S_{14}$ |
| Upper Crust Bohemia Volcanics | 15 | thermal conductivity | $\lambda_{14}$ |
| | | radiogenic heat production | $S_{14}$ |
| Upper Crust Saxothuringia | 16 | thermal conductivity | $\lambda_{16}$ |
| | | radiogenic heat production | $S_{16}$ |
| Upper Crust Vosges | 17 | thermal conductivity | $\lambda_{17}$ |
| | | radiogenic heat production | $S_{17}$ |
| Upper Crust Molasse | 18 | thermal conductivity | $\lambda_{18}$ |
| | | radiogenic heat production | $S_{18}$ |
| Upper Crust East Alps | 19 | thermal conductivity | $\lambda_{19}$ |
| | | radiogenic heat production | $S_{19}$ |
| **General-Focus Model:** | | | |
| Upper Crust West Alps | 20 | thermal conductivity | $\lambda_{20}$ |
| | | radiogenic heat production | $S_{20}$ |
| **Crustal-Focused Model:** | | | |
| Upper Crust West Jura | 20 | thermal conductivity | $\lambda_{20}$ |
| | | radiogenic heat production | $S_{20}$ |
| Upper Crust West Alps | 21 | thermal conductivity | $\lambda_{20}$ |
| | | radiogenic heat production | $S_{20}$ |
| Upper Crust Po Basin | 22 | thermal conductivity | $\lambda_{22}$ |
| | | radiogenic heat production | $S_{22}$ |
| Upper Crust North East Adria | 23 | thermal conductivity | $\lambda_{23}$ |
| | | radiogenic heat production | $S_{23}$ |
| Upper Crust Ivrea | 24 | thermal conductivity | $\lambda_{24}$ |
| | | radiogenic heat production | $S_{24}$ |
| Upper Crust South East Adria | 25 | thermal conductivity | $\lambda_{25}$ |
| | | radiogenic heat production | $S_{25}$ |
| Lower Crust Saxothuringia | 26 | thermal conductivity | $\lambda_{26}$ |





| | | radiogenic heat production | $S_{26}$ |
|---|---|---|---|
| Lower Crust Moldanubia | 27 | thermal conductivity | $\lambda_{27}$ |
| | | radiogenic heat production | $S_{27}$ |
| Lower Crust Alps | 28 | thermal conductivity | $\lambda_{28}$ |
| | | radiogenic heat production | $S_{28}$ |
| Lower Crust Ivrea | 29 | thermal conductivity | $\lambda_{29}$ |
| | | radiogenic heat production | $S_{29}$ |
| Lower Crust Liguria and Apennine | 30 | thermal conductivity | $\lambda_{30}$ |
| | | radiogenic heat production | $S_{30}$ |
| Lower Crust Adria | 31 | thermal conductivity | $\lambda_{31}$ |
| | | radiogenic heat production | $S_{31}$ |
| Lithospheric Mantle North West | 32 | thermal conductivity | $\lambda_{32}$ |
| | | radiogenic heat production | $S_{32}$ |
| Lithospheric Mantle South East | 33 | thermal conductivity | $\lambda_{33}$ |
| | | radiogenic heat production | $S_{33}$ |

*Author contributions.* All authors discussed and interpreted the presented work. DD carried out the simulations and all authors read and approved the final manuscript. CS generated the Crustal Focus Alps model.

*Competing interests.* The authors declare that they have no conflict of interest.

*Acknowledgements.* The authors gratefully acknowledge the Earth System Modelling Project (ESM) for funding this work by providing computing time on the ESM partition of the supercomputer JUWELS (Jülich Supercomputing Centre, 2019) at the Jülich Supercomputing Centre (JSC) under the application 16050 entitled "Qunatitative HPC Modelling of Sedimentary Basin System."



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
