# Peer review of "How biased are our models? - A Case Study of the Alpine Region"

_Geoscientific Model Development, 2021_

## Author Comment (AC1)

**Answer to the Report of Reviewer 1 of How biased are our models? – A Case Study of the Alpine Region**

Thank you very much for your report of our paper. Your remarks and comments
helped us to improve our paper. All changes are marked in red in the manuscript. Please, find below
a detailed answer to the individual remarks.

This study demonstrated that focusing analyses purely on measurements could introduces bias. This
research is performed based on a geothermal model, and Global Sensitivity Analysis and Reduced
Order Modeling methods are used to illustrate the influence of data distribution. The authors used a
weighting scheme to compensate for parts of this bias caused by data distribution. However, I think
the more important issue in this study is the general rule of a reliable weighting scheme. In addition,
some important details are not clear in this manuscript and should be addressed. Please see the
specific comments below.

1. Line 140 Are the weights determinted based on the number of data points? In addition, the
   author should describe how to use these weights briefly.

   The weighting scheme is determined by both the number of data points and information
   about the data quality (as described in lines 136-142). We applied the weighting scheme
   directly to the quantity of interest in all analyses. An explanation and the corresponding
   formula has been added in line 143-149.

2. Line 145 The two models have different geological layers (31 and 34), and why they have the
   same model space?

   The models have the same model space since the higher number of layers in the second
   model is obtained through a subdivision of several layers of the first model (as shown in
   Table A1). An explanation has been added to lines 157-158.

3. Line 178: what's the meaning of this threshold? the first- or total order indices?

   The threshold is applied for the total-order indices. An explanation has been added to line
   187.

4. Line 184: Before Figure 4, please explain all model parameters in a table.

   The model parameters are all presented in Table A1. To highlight that this paper has a
   method focus and not a focus on this specific case study, we placed the table on purpose in
   the Appendix since that has been a misunderstanding in previous publications.

5. Line 304: Again, how to use these weights? Are these weighs used for sensitivity analysis?
   Please give the formulas;

   An explanation and the corresponding formula has been added in lines 143-149 (see point
   1).

6. Line 316: It seems that the appropriate and reliable weighting scheme is crucial to compensate the data distribution problems. In addition, does the inappropriate weighting scheme may lead to bias to model output?

   Yes, an inappropriate weighting scheme does induce a significant bias to the results. We added this for clarification in lines 343 and 344.

7. Line 325: The authors should describe the RB surrogate models, e.g., the accuracy, the cost.

   We added Section 3.4 for a more detailed describe the RB surrogate models.

8. Line 340: Please give the results of model calibration, e.g., RMSE, R2.

   We added the R2 values in lines 371-374.

---

## Author Comment (AC2)

**Answer to the Report of Reviewer 2 of How biased are our models? – A Case Study of the Alpine Region**

Thank you very much for your report of our paper. Your remarks and comments
helped us to improve our paper. All changes are marked in red in the manuscript. Please, find below
a detailed answer to the individual remarks.

**General Comments:**

This paper is a well-designed and thoroughly referenced analysis of the biases introduced by
applying poorly distributed measurements to the assessment of geothermal models and of the
sensitivity of the influence of two model parameters (thermal conductivity and radiogenic heat
production) on the modeled subsurface temperature. The problem being addressed is adequately
described. However, it becomes difficult to determine if the authors have made their case
effectively, since the sequence of the evidence being presented is often confusing and the density of
the information discussed becomes rather high. For example, several figures are presented out of
numerical order and Figure 12 is not described in the text at all. There are 114 histogram boxes in
Figure 10 alone and 90 boxes in Figure 13. It is suggested that the manuscript would benefit and
would more clearly communicate its message if there were fewer figures, if the figures were
discussed in their numerical order, and if some of the figures were simplified. In addition, there are
numerous grammatical lapses that are detailed in the technical comments and must be addressed
before publication. The conclusions of this paper will be a worthwhile contribution to the literature,
but publication is recommended only after major revisions to the text and figures are completed.

**Specific (Scientific) Comments:**

**1 Introduction**

Page 2, Line 28: This section reads such that the first and second problems listed both relate to the
issue of data density with one directed in the vertical and the other related to horizontal data
scarcity. If an additional distinction was intended, please clarify.

Thank you for your comment. That was indeed the intended meaning. To clarify this, we rephrased
"related to density" to "also related to data density".

Page 2, Line 45: It might be helpful for the authors to clarify that by "global sensitivity analysis" they
are referring to examining the entire parameter domain within the spatial extent of their model and
are not referring to examining the parameters over the entire Earth.

We added a clarification on page 2 in line 44.

**2 Materials and Methods**

Page 3, Line 77: It's not clear what is meant by "…only the vicinity of the input parameters is
explored". Please clarify.

We clarified the meaning of "vicinity of the input parameters" on page 3 in line 78.

Page 5, Line 117: The data described are for the "southern foreland" of the Alps?

*Yes, the data is for the southern foreland of the Alps. We added a clarification on page 5 in line 118.*

Page 5, Line 130: Do the temperature databases provide some criteria or indicator of data quality, or if not, what basis or method was used to establish the data quality to inform the data filtering and weighting?

*Unfortunately, the database does not include an indicator of the data quality. Therefore, the weighting scheme is mainly based on the unequal data distribution. For the regions of the Upper Rhine Graben and Alps, we used an expertise-driven approach that is based on the extensive knowledge of the authors from previous studies in these regions.*

Page 6, Line 135: The number of measurements for the four regions adds to 2,391 data points, but the previous sentence identifies 2,388 total data points. Please explain or correct the difference.

*The difference has been corrected.*

Page 7, Line 142: The rationale for the weighting factor of 0.5 for the Upper Rhine Graben and Alps data points is not obvious. Please elaborate.

*As mentioned above this factor is determined based on the experience of the authors. The value can be updated if quantitative measures of the data quality become available. Note that the study aims to illustrate the effect of the bias induced by the measurements rather than providing an optimal weighting scheme. A clarification has been added to lines 146-149 (page 7).*

**3 Alpine Region**

Page 11, Line 217: What is the difference between Figures 6 and 9? In general, it is good practice in a manuscript to clearly describe each figure within the text and to introduce them in numerical order. Presenting the figures out of numerical sequence creates unnecessary confusion for the reader.

*The difference is that the analysis in Figure 6 focuses on the entirety of the model combining several layers with equal thermal properties, whereas the analysis of Figure 9 focuses on the Upper Crust layers only. Therefore, it further differentiates within the Upper Crust layers which are combined into one parameter in the analysis presented in Figure 6. A clarification has been added to lines 219-223 (page 9/10). Furthermore, we revised the entire manuscript to ensure that all Figures are introduced and mentioned in their numerical order.*

Page 11, Line 234: The sentence that begins "Note that we do not present the results..." is confusing on several points. First, the text should introduce what is presented in Figure 8 prior to this sentence (and prior to introducing Figure 9). Also, Figure 7 does present a sensitivity of thermal conductivity for the Upper Crust in the Po Basin, and Figure 8 does present a sensitivity of thermal conductivity for the Lower Crust in Saxothuringia, which conflict with the statement in the referenced sentence. Please clarify.

*That seems to be a misunderstanding. The results of Figures 7 and 8 are considered and presented in this section. However, we exclude the Upper crust property ($\lambda_{22}$ in Figure 7) and the Lower Crust property ($\lambda_{26}$ in Figure 8). These two properties are directly taken from the top-level analysis and required to ensure compatibility between the top- and low-level analyses. A clarification has been added to lines 253-256 (page 12).*
*Furthermore, as mentioned above we revised the manuscript with respect to the numerical order of the figures.*

Page 12, Line 251: The organization and description of the figures discussed in this section are confusing, and Figure 12 is not introduced in the text at all. The reader should be provided a clear motivation and description for each figure in the manuscript.

As mentioned above the Section has been revised with respect to the mentioning and ordering of the Figures. Furthermore, we added a clarification for Figure 12 on page 12 in line 264.

**4 Discussion**

Page 15, Line 280: The sentence that begins "This might be a bias introduced…" is confusing. The number of data points (755) is the same for the Unconsolidated Sediments in URG ($l_1$) and for the rest ($l_3$)?

That was a mistake, we meant $\lambda_1$ and not $\lambda_3$. We corrected and specified the sentence on page 17 in line 315.

Page 17, Line 317: It seems like an overstatement to claim that "Hence, the bias…can be removed". The weighting procedure is not creating new information, it's simply rebalancing existing information. It would be more appropriate to say that the bias can be "reduced", especially in the context of the later statement that the paper does not aim to provide the ideal weighting scheme for the Alpine Region.

We changed the term "removed" to "reduced".

**Tables and Figures**

Figure 6: What is the purpose of the thick horizontal black line? Remove if not needed. Same comment for Figures 10, 15, and 16.

The purpose of the black line is to indicate the threshold of what we consider as sensitive in this paper. An explanation has been added to the caption of Figure 6, 10, 15, and 16.

**Technical Corrections:**

All technical corrections regarding the text have been incorporated into the manuscript.

**Tables and Figures**

Figure 10: Due to the high information density, it's recommended to orient the boxes consistently (for example, hatched, colored, slanted) in all cases. This will make it easier for the reader to distinguish the sections of histograms. Also, this figure could be made somewhat wider to add some white space between the model parameter sections. Same comment for Figure 13.

The boxes have been updated and are now oriented consistently. Furthermore, the whitespace between the parameters has been increased. The updates have been performed for Figures 10-13.